# Testing the influence of environmental heterogeneity on fish species richness in two biogeographic provinces

Philippe Massicotte*, Raphaël Proulx, Gilbert Cabana and Marco A. Rodríguez

Centre de Recherche sur les Interactions Bassins Versants-Écosystèmes Aquatiques (RIVE), Université du Québec à Trois-Rivières, Trois-Rivières, Canada
* Current affiliation: Department of Bioscience, Aarhus University, Frederiksborgvej, Roskilde, Denmark

## ABSTRACT

Environmental homogenization in coastal ecosystems impacted by human activities may be an important factor explaining the observed decline in fish species richness. We used fish community data (>200 species) from extensive surveys conducted in two biogeographic provinces (extent >1,000 km) in North America to quantify the relationship between fish species richness and local (grain <10 km$^2$) environmental heterogeneity. Our analyses are based on samples collected at nearly 800 stations over a period of five years. We demonstrate that fish species richness in coastal ecosystems is associated locally with the spatial heterogeneity of environmental variables but not with their magnitude. The observed effect of heterogeneity on species richness was substantially greater than that generated by simulations from a random placement model of community assembly, indicating that the observed relationship is unlikely to arise from veil or sampling effects. Our results suggest that restoring or actively protecting areas of high habitat heterogeneity may be of great importance for slowing current trends of decreasing biodiversity in coastal ecosystems.

## INTRODUCTION

The environmental heterogeneity hypothesis (*MacArthur & MacArthur, 1961*; *MacArthur & Wilson, 1967*; *Ricklefs, 1977*) states that species richness increases with the number of ecological niches; that is, species coexistence is facilitated in more heterogeneous environment because different taxa can capitalize on different environmental conditions. The hypothesis has been tested using many taxonomic groups across different spatial grains (average distance among observations) and extents (size of the whole study area) ranging from meters to thousands of kilometers. An extensive meta-analysis by *Field et al. (2009)* found that environmental heterogeneity was the primary factor driving species richness for 63 of the 273 cases (23%) assessing the relative importance of environmental heterogeneity versus other environmental factors. Environmental heterogeneity, however, had a stronger effect on species richness in studies conducted at small grain sizes (39% of

Corresponding author
Philippe Massicotte, pm@bios.au.dk

the cases), suggesting that the relationship is contingent on the spatial scale. Furthermore, only 4 of the 393 relationships (1%) were from surveys of aquatic ecosystems having small grain size ($<10\,km^2$) and large geographical extent ($>1,000\,km$). Thus, there appears to be no consensus on the effects of small-grain environmental heterogeneity on species richness when investigated over large geographical areas.

This paucity of broad-scale studies may be related to the difficulties faced by aquatic ecologists in quantifying heterogeneity across different temporal and spatial scales (*Kovalenko, Thomaz & Warfe, 2011*; *Yeager, Layman & Allgeier, 2011*; *Tisseuil et al., 2013*) possibly reflecting the difficulties of obtaining the data needs to quantify such relationship. As a consequence, the term 'heterogeneity' has been used rather loosely, as it could refer to habitat complexity, habitat diversity or environmental variability in both space and time (*Palmer, Menninger & Bernhardt, 2010*). For example, *Oberdorff et al. (2011)* assessed habitat heterogeneity at the continental scale using the proportion of different biomes found within river drainage basins, whereas *Guégan, Lek & Oberdorff (1998)* used the mean annual flow discharge as a proxy for environmental heterogeneity in 183 rivers throughout the world. Although these two studies found a positive relationship between heterogeneity and fish species richness, their measures of environmental heterogeneity were confounded with biogeographic factors, such as the size of the drainage area and with other global environmental descriptors including seasonality of rainfall in lotic systems. Recent meta-analyses of the relationship concluded that environmental homogenization has a consistent and negative impact on animal diversity (*Smokorowski & Pratt, 2007*; *Seiferling, Proulx & Wirth, 2014*).

Empirical evidence of the relationship between local (small grain) fish species richness and environmental heterogeneity remains sparse for aquatic ecosystems of broad spatial extent. Further evaluation of the relationship is needed, especially when considering that: (1) species richness is declining in both freshwater and marine ecosystems (*Ricciardi & Rasmussen, 1999*; *Worm et al., 2006*), and (2) aquatic ecosystems are increasingly impacted by human activities, such as systematic embankment, river damming and seafloor trawling that are causing environmental homogenization (*Lotze et al., 2005*; *Jackson, 2008*) as well as changes in water quality variables (*Rabalais, 2002*). Fish communities are affected by structural characteristics of the environment such as reef structure and the presence of vegetation (*Kuffner et al., 2006*) and also by water quality variables such as salinity, turbidity, and oxygen concentration (*Rabalais, 2002*; *Bejarano & Appeldoorn, 2013*). The objective of this study was to evaluate the effect of local environmental heterogeneity in environmental variables (spatial grain $<10\,km^2$) on fish species richness at the scale of biogeographic regions (spatial extent $>1,000\,km$). We used data on fish communities (26 orders, 73 families, 136 genera, 204 species), obtained from extensive surveys in two coastal ecosystems of North America. Using a set of environmental variables routinely measured by monitoring programs, we demonstrate that fish species richness in coastal ecosystems responds positively to the spatial heterogeneity of environmental conditions and quantify the magnitude of this effect. We implemented a random placement model of community

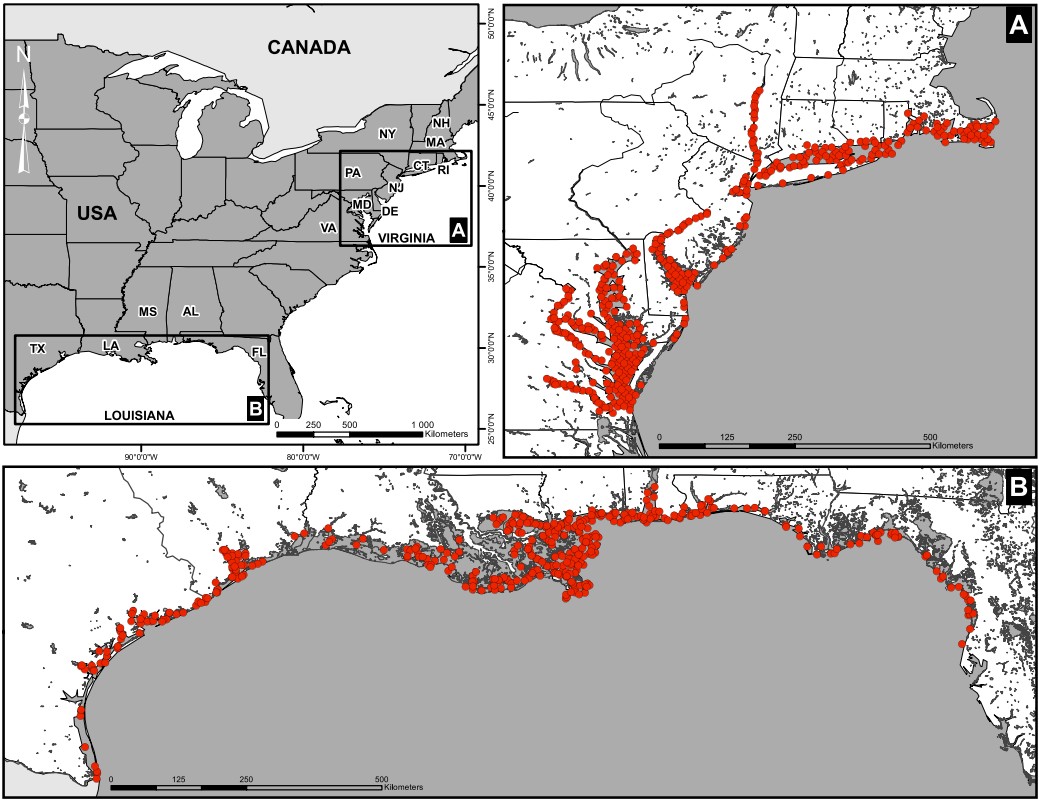

**Figure 1 Spatial distribution of sampling sites.** (A) Virginian and (B) Louisianan biogeographic provinces. Surveys were conducted by the U.S. Environmental Protection Agency's Environmental Monitoring and Assessment Program (EMAP) between 1990 and 1994.

assembly to ensure that the empirical relationship found between species richness and environmental heterogeneity did not result of a veil or sampling effect.

## MATERIAL AND METHODS

### Study site and data collection

Fish abundances and environmental measurements were obtained from two extensive surveys conducted by the U.S. Environmental Protection Agency's Environmental Monitoring and Assessment Program (EMAP). The first data set consisted of four sampling campaigns conducted in the Virginian biogeographic province between 1990 and 1993 (*Hale et al., 2002*). Stations were located along the coastline and in large river estuaries of the East Coast (Delaware, Hudson, Potomac, York; Fig. 1A). The second data set was assembled from four sampling campaigns conducted in the Louisianan biogeographic province between 1991 and 1994. Stations were located along the Gulf of Mexico from the Rio Grande, Texas, to Anclote Island, Florida (Fig. 1B). Field campaigns in the two biogeographic provinces were carried out between July and September of each year.

Fish were sampled using balloon trawls (funnel-shaped nets, 4.9 m wide with 2.5 cm stretched mesh) deployed from a research vessel using a hydraulic–powered boom in the vicinity of the sampling stations. The duration of the trawl was $10 \pm 2$ (mean $\pm$ SD) minutes at a speed of 2–3 knots. This corresponds to a length of $0.77 \pm 0.15$ (mean $\pm$ SD) km. Following a successful trawl, the net was hauled aboard and the catch was released into a plastic trough, or a fish sorting table, where species composition and abundance were recorded (see Appendix S1). A total of 2,237 individuals (fork length: min. $= 2.2$ cm; max. $= 91.18$ cm; mean $\pm$ SD $= 12.08 \pm 7.33$ cm) were captured from the Louisianan biogeographic province and 1,883 individuals (fork length: min. $= 2.5$ cm; max. $= 92.6$ cm; mean $\pm$ SD $= 16.03 \pm 10.37$ cm) were captured from the Virginian biogeographic province, yielding a total of 4,120 individuals (Table 1, Appendix S1).

The environmental data comprised physical and chemical measurements. Dissolved oxygen concentrations (mg $\times$ L$^{-1}$) were determined using an air-calibrated oxygen meter (Yellow Springs Instruments, Yellow Springs, OH, USA) on surface water samples (625 mL) obtained with a Go-Flo bottle (General Oceanics, Miami, Florida, USA). Salinity (ppt), temperature (°C), pH, transmissivity (% of ambient light transmitted through the water column), photosynthetically active radiation ($\mu$E $\times$ m$^{-2}$ $\times$ s$^{-1}$), fluorescence (unitless) and water density ($\sigma$t, kg $\times$ m$^{-3}$ $- 1,000$) were measured using a SeaBird CTD meter (Sea-Bird Electronics, Bellevue, Washington, USA) lowered through the water column at a rate of approximately 0.25 m $\times$ s$^{-1}$ until it reached the bottom (Table 1). Fluorescence and water density data were not available for the Louisianan surveys. Implicit to our approach is that the gradient of environmental conditions captures a range of habitat and resource types. For example, temperature and dissolved oxygen may to some degree correlate with water depth or nutrient loading, whereas the photosynthetically active radiation is a more direct measure of primary production in the water column. Other variables, such as salinity, impose physiological constraints to the distribution of fish species in coastal transition zones. Detailed information about the sampling and analytical procedures can be found on the EMAP web site (http://www.epa.gov/emap/index.html). Although other environmental variables such as macrophyte cover might be important determinants of environmental heterogeneity, the selected variables are known to affect the ecology of individual fish species (*Mandrak, 1995*).

### Environmental heterogeneity

To represent the gradient of environmental conditions among stations of the same biogeographic province, we used the scores of a principal component analysis (PCA) performed on the environmental variables. The first three PCA axes (Table 1) were retained based on Kaiser's criterion (*Kaiser, 1960*) and explained nearly 75% of the environmental variability in both Virginian (PC1 $= 42.28\%$, PC2 $= 19.7\%$, PC3 $= 12.6\%$) and Louisianan (PC1 $= 32.5\%$, PC2 $= 23.8\%$, PC3 $= 19.6\%$) biogeographic provinces. We quantified the degree of local spatial autocorrelation in environmental conditions near each station as a reciprocal measure of environmental heterogeneity.

Massicotte et al. (2015), *PeerJ*, DOI 10.7717/peerj.760

**Table 1 Loadings and summary statistics for environmental variables.** The first three principal components generated from environmental variables were retained based on Kaiser's criterion. These components explained 75% of the total environmental variability in both biogeographic provinces.

| Variable | Virginian | | | | | | Louisianan | | | | | |
|---|---|---|---|---|---|---|---|---|---|---|---|---|
| | Loadings | | | | | | Loadings | | | | | |
| | Comp. 1 | Comp. 2 | Comp. 3 | Mean | Std. Dev. | Range (min–max) | Comp. 1 | Comp. 2 | Comp. 3 | Mean | Std. Dev. | Range (min–max) |
| Water density ($\sigma_t$) | −0.49 | 0.02 | 0.12 | 9.08 | 8.68 | −4.36–23.94 | | | | | | |
| Dissolved oxygen (mg L$^{-1}$) | −0.10 | −0.69 | 0.03 | 6.90 | 1.25 | 3.0–11.2 | −0.42 | 0.55 | −0.10 | 6.89 | 1.33 | 3.4–14.8 |
| Fluorescence | 0.28 | −0.34 | 0.42 | 11.82 | 7.70 | 0–30 | | | | | | |
| PAR ($\mu^{-2}s^{-1}$) | −0.05 | −0.27 | −0.85 | 545.76 | 464.29 | 9–3621 | −0.51 | −0.41 | −0.10 | 813.25 | 477.61 | 12–1,870 |
| pH | −0.28 | −0.53 | 0.16 | 7.93 | 0.48 | 6.3–9.4 | −0.40 | 0.47 | 0.41 | 8.00 | 0.46 | 5.3–9.5 |
| Salinity (ppt) | −0.49 | 0.00 | 0.11 | 16.18 | 11.05 | 0.03–32.89 | −0.06 | −0.14 | 0.84 | 13.47 | 10.70 | 0.01–37.35 |
| Temperature (°C) | 0.39 | −0.21 | −0.16 | 25.40 | 2.46 | 11.80–30.85 | −0.50 | 0.02 | −0.32 | 29.77 | 1.41 | 24.7–34.0 |
| Transmissivity (%) | −0.44 | 0.10 | −0.14 | 53.37 | 23.19 | 0–93 | −0.39 | −0.54 | 0.11 | 63.97 | 16.12 | 2–133 |

We calculated the local Moran $I$ statistic on the scores of the first three PCA axes using the `localmoran` function of the `spdep` package in R (*Bivand et al., 2013*; *Bivand & PIras, 2015*). Only the first PCA axis was retained for further analysis because we did not find any relationship between Moran's $I$ calculated for PCA axes 2 or 3 and species richness. The Moran $I$ statistic identifies station neighborhoods where environmental conditions of similarly high or low values cluster spatially (high $I$), as well as neighborhoods where environmental conditions are more contrasted (low $I$). High $I$ values indicate low heterogeneity (positive autocorrelation), whereas values around zero indicate high heterogeneity. Negative $I$ values indicate local over-dispersion patterns (i.e., negative autocorrelation), which are rarely observed in nature (*Borcard, Gillet & Legendre, 2011*). The $I$ statistic is given by *Anselin (1995)*:

$$I = (n-1) \frac{x_i - \bar{X}}{\sum\limits_{i=1}^{n}(x_i - \bar{X})^2} \sum\limits_{j=1}^{n} w_{ij}(x_j - \bar{X}) \tag{1}$$

where $x_i$ is the value of the observation $i$, $\bar{X}$ is the mean of the variable, $w_{ij}$ is the spatial weight ($1/distance^2$) between observations $i$ and $j$, and $n$ is the number of stations sampled. Each $I$ (one per station) have been calculated by including all surrounding neighbours in a 75 km radius using the `dnearneigh` function of the `spdep` package. The chosen radius was large enough to include a sufficient number of neighboring stations in the calculation of Moran's $I$ (average number of stations: Virginian = 54.3; Louisianan = 52.7), while small enough to prevent the inclusion of stations that are only remotely connected. Spatial weights were scaled as $1/distance^2$ in Eq. (1), thus varying the search radius had little effect on $I$ values. Because we could not determine whether patterns of over-dispersion should be associated with high or low levels of environmental heterogeneity, the few stations (less than 4%) with negative $I$ values were removed from subsequent statistical analyses. We did not find substantial differences between results for $I$ calculated using all the data pooled at the biogeographic level (spatio-temporal $I$) and $I$ calculated for each sampling year separately (spatial $I$). Consequently, we view $I$ as a measure of spatial heterogeneity in local environmental conditions across space (Appendix S2, Fig. 1, Eq. (1)).

## Numerical simulations

We developed a random placement model of community assembly to determine the heterogeneity–species richness relationship in the absence of explicit habitat selection mechanisms. The model has two main components: (1) environmental heterogeneity and (2) species richness, each being simulated independently of the other on a two-dimensional surface (Fig. 2). This approach has been successfully used in various ecological studies aiming to highlight the effect of landscape structures on different aspects of animal biodiversity (*McGill, 2011*; *Campos et al., 2013*).

The first model component simulates the spatial patterns of environmental conditions (Fig. 2A). Environmental spatial patterns can be modeled as a fractional Brownian

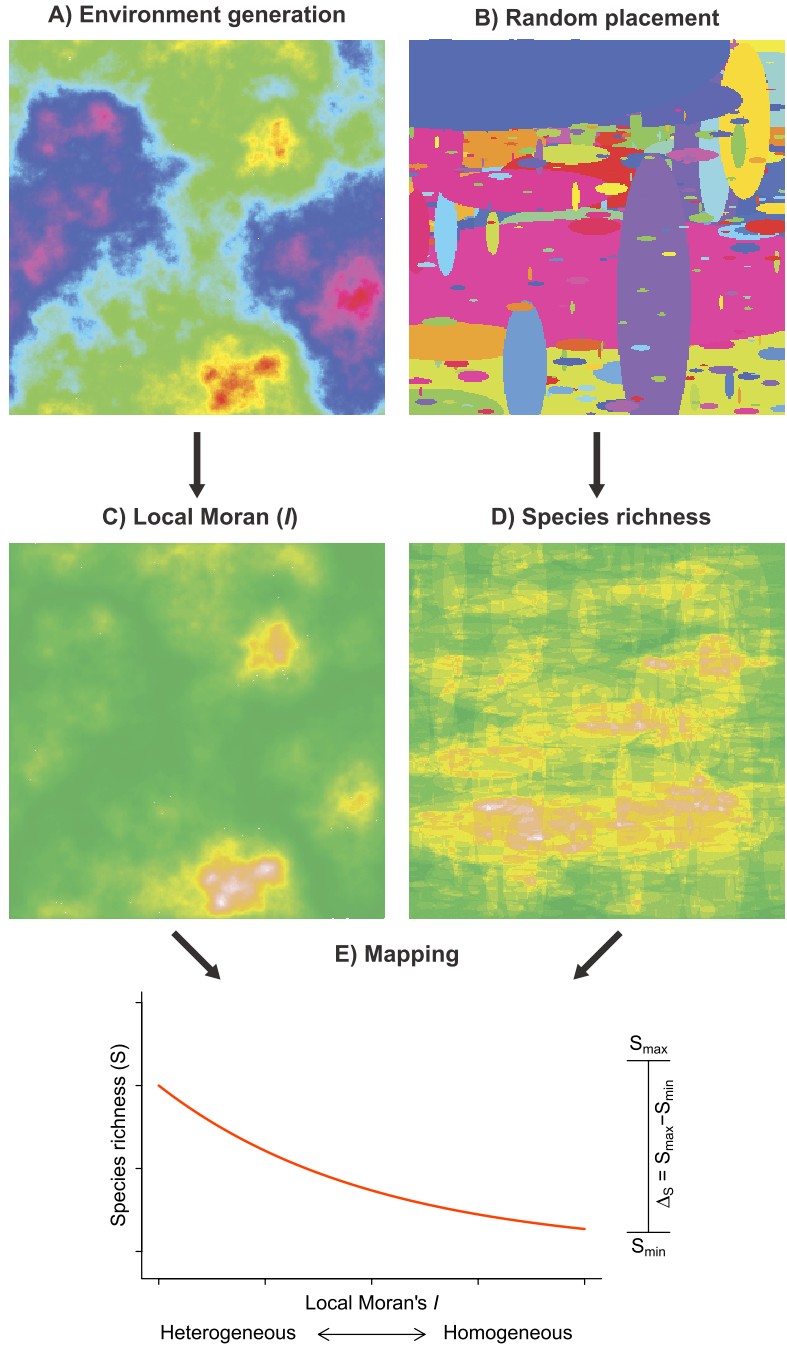

**Figure 2 Framework of the random placement model of community assembly used to determine the relationship between fish species richness (S) and habitat heterogeneity in the absence of any particular habitat selection mechanisms.** Both environmental scores (A) and the species spatial distributions (B) were generated independently and parameterized using observed data. Habitat heterogeneity (C) and species richness (D), the two resulting model components, were superimposed such that each Moran's $I$ value on the grid was associated to a value of species richness (E). $S_{min}$ and $S_{max}$ represent the range spanned by a fitted GLM negative binomial regression (red curve). To simulate possible artifacts due to unsampled fish (false 0), we added a veil effect threshold to the data generated by the model. A total of 10,000 simulation were produced.

function. The spectral density $S(f)$ of a two-dimensional surface follows a power spectrum $S(f) \propto 1/f^{\beta}$ (*Keitt, 2000*), where $f$ is frequency and $\beta = 1 + 2H$. The Hurst exponent ($H$) controls the degree of auto-correlation in environmental conditions; a large $H(H \longrightarrow 1)$ results in relatively homogeneous spatial patterns, whereas a lower $H(H \longrightarrow 0)$ produces more heterogeneous patterns. To generate the environmental spatial patterns in our simulations, we used the Matlab function `noiseonf`, which uses the inverse Fourier transformation of a power spectrum with a predetermined Hurst exponent (*Kovesi, 2000*). This procedure generates 'neutral' landscapes (e.g., *With, 1997*; *Keitt, 2000*) that share several statistical properties with environmental patterns observed in nature. The Hurst exponent of the simulated surface was parameterized using the linear slope of the log–log semi-variogram (*Gallant et al., 1994*) computed on the scores of the first axis of the PCA of environmental conditions, yielding values of $H \approx 0.4$ in both biogeographical provinces.

The second component (Fig. 2B) of our model simulates the random placement of species with different distribution ranges. We based our random placement model of community assembly on two premises (*McGill & Collins, 2003*; *McGill, 2010*): (1) the centroid of each species range is determined by sampling from a uniform distribution over the surface and (2) the range size of species is distributed according to a power distribution. *McGill & Collins (2003)* reported that implementing either a log-normal or a power distribution did not affect the results of random placement model. Each of our simulation runs proceeded as described in Algorithm 1. Local species richness is then calculated by summing the overlap of different species ranges. On the basis of the observed regional distributions of the sampled species (Appendix S2, Fig. 3), we used the following parameters to implement the random placement model: $G = 1,000$, $r_{min} = 10$ km and $r_{max} = 1,000$ km.

---

Algorithm 1. Random placement of species (component 1, Fig. 2A)

1. Generate a surface of size $G \times G$.
2. Randomly pick the distribution range $r$ of a new species from a power function
   $f(r) = r^{-a}$ where $r_{min} \leq r \leq r_{max}$ (Appendix S2, Fig. 2).
3. Choose the species centroid randomly from a uniform distribution over the surface.
4. Repeat previous steps until the surface is completely covered by species ranges (ranges are allowed to overlap).

---

To represent the range of each species on the surface, we used ellipses with major axis length $r$ and minor axis length sampled from a uniform in the interval $[r/4, r/2]$ as described in *Proulx et al. (2014)*. To simulate an anisotropic spatial process, we placed the elliptical ranges with their major axis oriented either horizontally (with probability = 0.75) or vertically (with probability = 0.25). This decision was motivated by the fact that species ranges in both biogeographical provinces are preferentially oriented along rivers and coastlines that broadly conform to the proposed alignment. Finally, to determine the parameter $\alpha$ empirically, we calculated the range of all fish species in each biogeographical province (Appendix S2, Fig. 3) and estimated the power coefficient of the frequency using the log-ratio formula (Eq. (5) in *Newman, 2005*). We obtained values of $\alpha = 1.214$ for

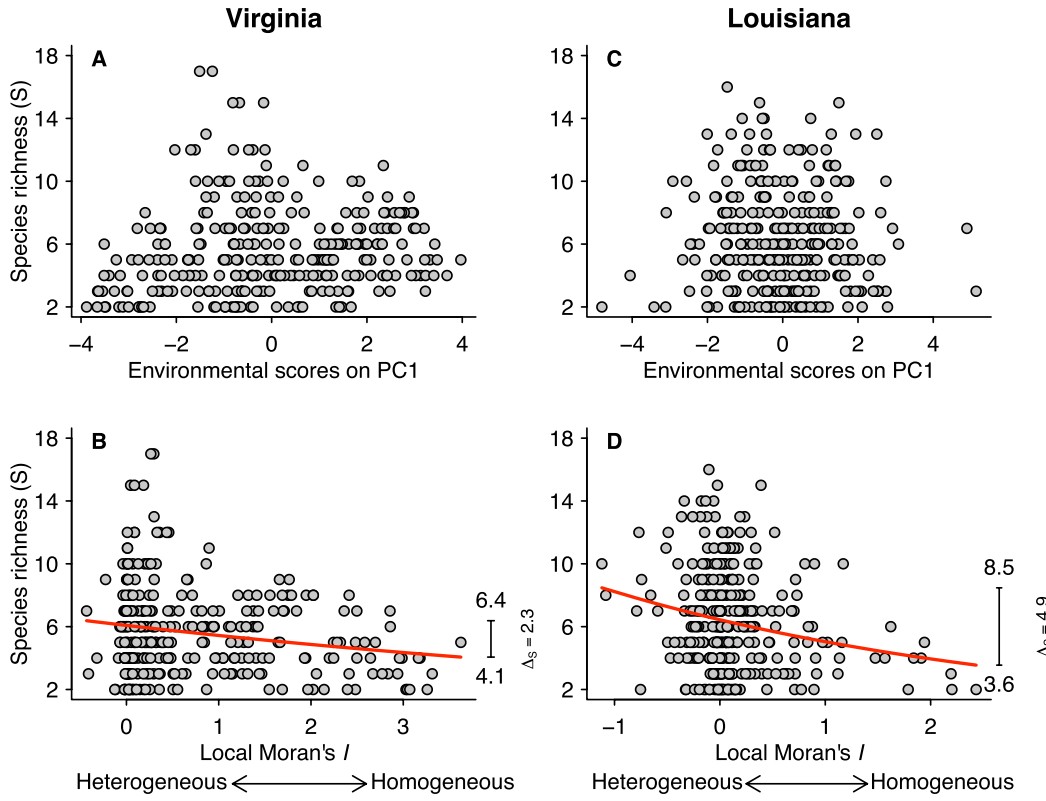

**Figure 3** **Relationships between species richness (*S*) and PCA scores for the first axis (A and C) and local Moran's *I* (B and D) for the Virginian and Louisianan biogeographic provinces.** The red lines represent the fitted GLM negative binomial regressions between local Moran's *I* and *S* (Virginian $p < 0.001$, Louisianan $p < 0.001$). The right-margin insets in (B) and (D) show the amplitude of species richness ($\Delta_S$) described by the regression curves.

the Virginian province and $\alpha = 1.189$ for the Louisianan province, and therefore used a value of 1.2 in our simulations. Using different combinations of ellipse shape ratio and orientation, we found that the species richness was robust to these changes. Most importantly, varying the shape ratio and orientation of the ellipse (species range) did not affect the general direction and relative effect size of the simulated environmental heterogeneity–species richness relationship. We generated the two model components on grids of $1,000 \times 1,000$ cells (Figs. 2A and 2B). A total of 10,000 simulations where performed according to Algorithm 2. It is to be noted that the model does not aim to approximate the absolute number of species at each location. Consequently, we used relative changes in species richness ($\Delta_S$) to compare modeled and observed results.

In each of the biogeographic provinces surveyed, approximately 5% of the stations yielded species richness values of zero. These zeros may partly arise from a 'veil effect' (*Preston, 1948*), and so reflect insufficient sampling effort rather than true absences. Truncation of samples at the veil may induce a spurious negative relationship between richness and predictor variables (Fig. 2E). To represent this effect in the simulated data,

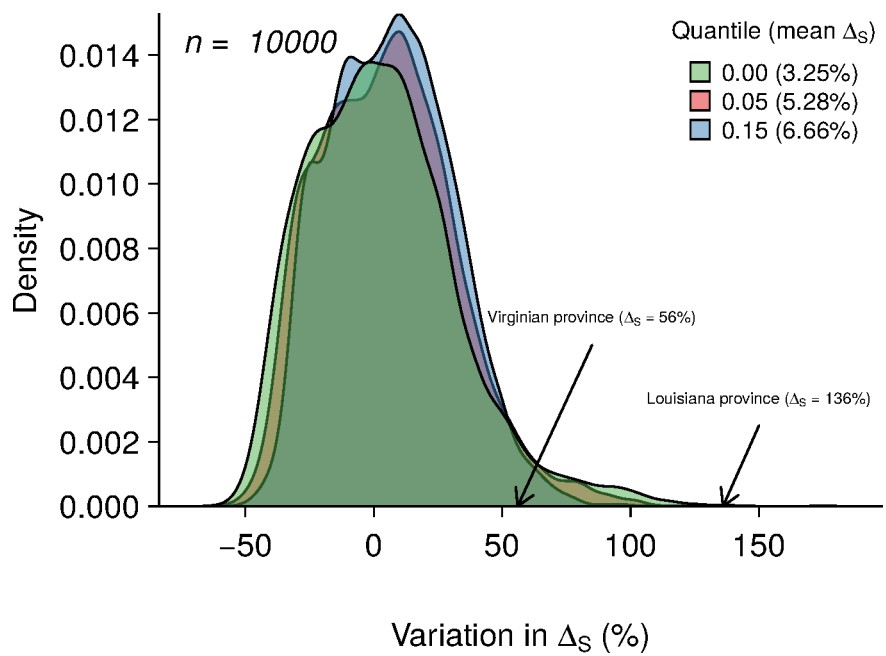

**Figure 4 Results of 10,000 simulations showing the influence of quantile cut (veil effect) on modeled species richness.** The green, red and blue areas represent the distribution of $\Delta_S$ under veil effects of percentiles 0%, 5% and 15%. The numbers in parentheses represent the mean of $\Delta_S$ for each veil simulation. The arrows indicate the $\Delta_S$ observed in the two biogeographic provinces.

we set three veil lines at percentiles 0%, 5% and 15% and excluded species richness values below these thresholds (Fig. 4).

Algorithm 2. Global simulation procedure
1. Generate an environmental grid (component 1, Fig. 2A).
2. Generate a species placement grid (component 2, Fig. 2B).
3. Randomly subsample 400 grid cells (roughly corresponding to the
    total number of sampling stations in each biogeographic province, Appendix S2, Fig. 4).
4. Calculate the local Moran's $I$ at each subsampled cell on the environmental grid following the procedure
    described in the *Environmental heterogeneity* section (Eq. (1), Appendix S2, Fig. 3).
5. Pair each local $I$ value to its associated species richness value on the environmental
    and the species placement grid, respectively.
6. Fit a negative binomial regression between the paired values of local Moran's $I$ and species richness
    (Fig. 2E).
7. Calculate the relative increase in species richness ($\Delta_S$) predicted by the regression curve.

## Statistical analyses

We used regression analyses to examine the relationships between species richness and the scores from the first PCA axis of environmental variables. To determine whether environmental heterogeneity had an influence on species diversity for both observed and simulated data, negative binomial regressions were fitted to the points above the veil effect

**Table 2 The probabilities of observing $\Delta_S$ greater or equal than 56% (Virginian) or 136% (Louisianan) due to sampling effect (i.e., random) under different scenarios of veil effects (0%, 5%, 15%).** See 'Methods' and Fig. 4 for detailed information.

|                    | Veil at 0% | Veil at 5% | Veil at 15% |
|--------------------|------------|------------|-------------|
| Virginian (56%)    | 4.68       | 3.70       | 2.12        |
| Louisianan (136%)  | 0.05       | 0.01       | 0.00        |

threshold using the `glm.nb` function of the `MASS` package in R (version 3.0.1). We also checked for the presence of spatial autocorrelation in the model residuals.

## RESULTS

Fish species richness was not correlated with any of the first three principal components from the analysis of environmental variables (Table 1; Figs. 3A and 3C), or with any of the individual environmental variables (results not shown). However, species richness was related to environmental heterogeneity (Figs. 3B and 3D). For both biogeographic provinces, the negative binomial regressions showed that species richness was greater in more heterogeneous environments (Figs. 3B and 3D). In the Virginian province (Fig. 3B), the mean species richness increased from 4.1 in the most homogeneous environments to 6.4 in the most heterogeneous environments, representing a gain of 2.3 ± 0.11 (95% confidence limits) species which correspond to 56% relative increase. A similar pattern was found for the Louisianan province (Fig. 3D) where mean species richness increased from 3.6 in the most homogeneous environments to 8.5 in the most heterogeneous environments, representing a gain of 4.9 ± 0.16 (95% confidence limits) species which correspond to 136% relative increase. We did not find spatial autocorrelation in the model residuals.

Averaging the results of 10,000 model simulations, the mean species richness relative increase ($\Delta_S$) were of 3.25%, 5.28% and 6.66% for the 0%, 5% and 15% veil effects, respectively (Fig. 4). The probabilities of observing $\Delta_S$ greater or equal to 56% (Virginian province) due to a sampling effect for different veils (0%, 5%, 15%) were of 4.68%, 3.7% and 2.12%, respectively (Table 2). Considering a $\Delta_S$ of 136% threshold (Louisianan province), these probabilities dropped to 0.05%, 0.01% and 0% (Table 2).

## DISCUSSION

Many factors, including environmental heterogeneity, have been reported to affect the diversity of aquatic communities (*Field et al., 2009*). However, it is likely that the set of factors influencing species richness differs across spatial and temporal scales (*Fausch et al., 2002*). Moreover, environmental heterogeneity has been identified as a key factor maintaining animal biodiversity in aquatic ecosystems (*Levin et al., 2010*). Our study combines data from extensive surveys and simulations to demonstrate a strong positive influence of environmental heterogeneity on the species richness of fish communities. Interestingly, species richness was associated with the spatial heterogeneity of environmental variables but not with their magnitude. For both biogeographic provinces, mean

species richness in the most heterogeneous environments was markedly greater than in the most homogeneous environments, as quantified by the negative binomial regressions. Furthermore, the observed effect of heterogeneity on species richness was substantially greater (Fig. 3) than that generated by the simulations based on a random community assembly model, so it seems unlikely that the observed relationship arose solely as a byproduct of veil or sampling effects.

### Environmental variables

Climate, energy, and primary productivity have a major influence on species richness at the regional, continental and global scales (*Guégan, Lek & Oberdorff, 1998*; *Hawkins, Field & Cornell, 2003*; *Field et al., 2009*). Studies conducted at small grain also indicate that environmental variables can influence the occurrence of species and abundance in local fish communities in both space and time (*Menge & Olson, 1990*; *Thiel et al., 1995*; *Rodríguez & Lewis, 1997*). In contrast to these findings, we did not observe any direct effect of the magnitude of individual environmental conditions (Table 1), including salinity, chlorophyll-*a* concentration and water temperature, on the species richness of local fish communities in either the Virginian (Fig. 3A) or Louisianan (Fig. 3C) biogeographic provinces.

In contrast with a simple randomization procedure, the simulation approach used in the random placement models allowed us to make a number of assumptions regarding the ecology of coastal fishes: (1) the spatial pattern of environmental conditions follows a two-dimensional power spectrum; (2) the centroid of each species on the seascape is determined by sampling from a uniform distribution and its range size by sampling from a power distribution; (3) fish species richness is independent from environmental conditions at the site of capture. We note that all three assumptions were supported by the empirical data. Another major assumption of random placement models is that the probability of finding a fish species at a particular site is independent of other species. Such ecological independence between co-occurring species has been shown to accurately reproduce a number of community patterns (*McGill, 2010*; *McGill, 2011*). For example, a recent study of shrubland plant communities reported that only 7–19% of all species pairs showed strong and consistent spatial associations, leading the authors to conclude that ecological processes are left no discernible spatial signature (*Perry et al., 2014*). In contrast with these findings, our results suggest that coastal fish communities may show a spatial signature, as fish species richness was not associated locally with the magnitude of environmental variables, but rather with their spatial heterogeneity.

### Environmental heterogeneity

Environmental heterogeneity influences many ecological processes such as fluxes of organisms, material and energy among riverscape elements (*Pickett & Cadenasso, 1995*). Our results demonstrate that fish species richness responded positively to increased environmental heterogeneity (Figs. 3B and 3D) in both the Virginian and Louisianan biogeographic provinces. Simulations using a random placement model of community assembly showed that species richness increased only slightly in more heterogeneous

environments (Fig. 4). For instance, less than 5% of the 10,000 simulations generated $\Delta_S$ greater than the conservative value of 56% observed in the Virgina biogeographic province (Figs. 3 and 4, Table 2). Hence, it is unlikely that the positive relationship observed between environmental heterogeneity and species richness in both biogeographic provinces is the result of a sampling effect (*sensu McGill, 2011*).

Aquatic ecologists often use the term 'heterogeneity' rather loosely to refer to habitat complexity, habitat diversity or environmental variability over time (reviewed in *Palmer, Menninger & Bernhardt, 2010*). For example, at small scales, heterogeneity usually refers to the variability in structural physical properties of the aquatic habitat such as riparian vegetation, channel configuration, artificial riffles and substrate granulometry (*Palmer, Menninger & Bernhardt, 2010*). Conversely, studies conducted at regional or continental scales have used large-grained variables such as percentage of different types of biome or drainage area as a proxy for habitat heterogeneity (*Guégan, Lek & Oberdorff, 1998*; *Field et al., 2009*; *Oberdorff et al., 2011*), possibly reflecting the difficulty of obtaining information at a finer resolution. Consequently, studies conducted at regional or continental scales are likely to capture broad-scale environmental heterogeneity that is coarse relative to the local heterogeneity to which individual fish respond, particularly for species having ranges smaller than the study grain size (*O'Neill et al., 1986*; *Turner et al., 1989*; *Wiens, 1989*). However, the question of which scale is optimal for quantifying the heterogeneity-diversity relationship is still open (*Chase & Leibold, 2002*; *Durance, Lepichon & Ormerod, 2006*; *Pittman et al., 2007*).

### Conclusions

Over the last century, coastal ecosystems have become increasingly impacted by anthropogenic pressures (*Lotze et al., 2006*), including many human–driven activities that reduce the temporal and spatial heterogeneity of coastal habitats. For example, commercial fish trawlers are known to reduce the spatial heterogeneity of the sea floor structure (*Helfman, 2007*). Similarly, the temporal variability of water flows in many of the world's largest rivers are regulated by dams (*Nilsson et al., 2005*). This reduced variability in runoffs has been shown to increase the homogeneity of water channels, as well as to degrade fish habitats (see *Moyle & Mount, 2007* and references therein). The current study shows that, independently of the environmental conditions prevailing locally, more homogeneous habitats can support fewer fish species. Hence, restoring or actively protecting areas of high habitat heterogeneity appears to be of great importance for slowing actual trends of decreasing biodiversity in coastal ecosystems.

### ACKNOWLEDGEMENTS

We thank the U.S. Environmental Protection Agency's Environmental Monitoring and Assessment Program (EMAP) for freely providing their data. Although the data described in this article have been funded wholly or in part by the U.S. Environmental Protection Agency through its EMAP Estuaries Program, it has not been subjected to Agency review, and therefore does not necessarily reflect the views of the Agency and no official

endorsement should be inferred. K Roach made helpful comments on an earlier version of the manuscript.

### Funding

Support was given by the Groupe de Recherche Interuniversitaire en Limnologie et Environnement Aquatique (GRIL). P Massicotte was supported by a postdoctoral fellowship from RIVE. The funders had no role in study design, data collection and analysis, decision to publish, or preparation of the manuscript.

### Grant Disclosures

The following grant information was disclosed by the authors:
Groupe de Recherche Interuniversitaire en Limnologie et Environnement Aquatique.
RIVE.

### Competing Interests

The authors declare there are no competing interests.

### Author Contributions

- Philippe Massicotte conceived and designed the experiments, performed the experiments, analyzed the data, wrote the paper, prepared figures and/or tables, reviewed drafts of the paper.
- Raphaël Proulx conceived and designed the experiments, performed the experiments, analyzed the data, wrote the paper, reviewed drafts of the paper.
- Gilbert Cabana and Marco A. Rodríguez conceived and designed the experiments, wrote the paper, reviewed drafts of the paper.

### Supplemental Information

Supplemental information for this article can be found online at http://dx.doi.org/10.7717/peerj.760#supplemental-information.

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
