# Peer review of "Testing the influence of environmental heterogeneity on fish species richness in two biogeographic provinces"

_PeerJ, doi:10.7717/peerj.760_

## Round 0.1 · original submission · Major Revisions

I note the mixed comments from the referees. I found the paper well written, and correctly points out the confusing use of terminology about heterogeneity in the literature. Such terms are very context dependent (I attempted to distinguish some in a paper on habitat classifications in MEPS in 2006). I think that there is an opportunity here to be more explicit about how the present data can clarify this. For example, the environmental variables used are not near ideal 'habitat' descriptors for fish species (perhaps for plankton). They are characteristics of the water whereas most fish species are benthic, demersal or associated with topographic features. The paper should make it clear how it has not included all physical habitat characteristics. For example, even depth, and wave and current exposure, appear to have been omitted. I think it essential to clarify how the sampled variables may actually affect species richness. to do this the reader needs to see the range of the variables. For example a 28-34ppt salinity will be fine for marine species and only at <26ppt may start selecting for euryhaline species. Similarly, if oxygen is always sufficient then it will not influence distributions and low oxygen excludes species. Thus variability (hetergeneity) in these variables seems unlikely to increase species richness on theoretical grounds. Really, the study should have looked at topographic variation and (related) wave and current exposure. Importantly, the paper needs to distinguish between 'environmental heterogeneity' (as measured here and indirectly may relate to species richness) and 'habitat' heterogeneity (which by definition habitat is directly associated with species' presence, so more habitats mean more species). In addition tot eh very abstract plots of statistics here, I think the paper should include plots showing the range of each environmental variable in relation to number of species at those sampling sites.

Reviewer 1 ·

Basic reporting

The manuscript is generally well written, and deals with a question that has been receiving quite a bit of attention recently: the relationship between species richness and habitat heterogeneity. The manuscript is unique in that it addresses this question in a marine setting, which compared to terrestrial systems has been infrequently studied. I have some comments on the presentation of the study.

The motivation behind this study (lines 30-34) seems unclear at first since the authors do not mention that human activities are causing homogenization of marine ecosystems (which subsequently can lead do diversity declines). This issue is only mentioned in the conclusion section (lines 226-231). I suggest that the authors describe this problem earlier, in the motivation paragraph (lines 30-34).

The first mention of the random placement model (lines 40-42) seems out of context, as it is unclear at this stage why it is needed. This is also the case at the beginning of the ‘Numerical simulations’ section (lines 99-100), where I struggled at first to understand why this analysis is needed. It is only in the discussion section (lines 211-212) that the authors state that the simulation model is needed to ensure that the empirically – found positive richness – heterogeneity relationship is not the outcome of sampling effects. Please state the reasoning behind the inclusion of this analysis when you first mention it in line 40.

Specific comments:

Line 78 – add reference to Kaiser’s criterion. Readers may be unfamiliar with it.

Lines 81–84 – It’s slightly unclear at which scale you’re calculating Moran’s I. I assume that it’s a station-scale measure, based on environmental data from all stations within 75 km of it. If this is the case, please state it explicitly.

Line 89 – X term missing before ‘is the mean of the variable’.

Line 180 – Are you sure that the scale of your analysis fits the scale at which species perceive and respond to their environment? If so, please add a reference to support this claim.

Figure 2 caption – I don’t see a dashed horizontal red line.

Experimental design

The experimental design is suitable for the research question, but I have some reservations about specific approaches.

Most importantly, isn’t there a discrepancy between the scale of richness data (individual station) and the heterogeneity data (a 75km buffer around the station)? To test the richness – heterogeneity relationship, I’d rather compare station-scale measures of both. I’m aware that you did not have station-scale data as each station had a single sample of environmental conditions. Also, understand that it’s reasonable to quantify heterogeneity at a slightly larger extent than the extent of the richness sample. Yet it is possible that you underestimated richness levels since you quantified richness based on a 770m transect, whereas you quantified heterogeneity for a circular area approaching 150km in diameter. Given that you don’t have the data to quantify heterogeneity at finer scales, I don’t suggest that you revise the approach, but it would be helpful if you addressed the issue of scale discrepancy in the discussion.

I understand why you wanted to ensure that the richness – heterogeneity relationship that you found is not an outcome of sampling effects. However, wouldn’t a standard null-model approach suffice to test for the presence of sampling effects? For a recent example, please see Stegen et al. (2013) Stochastic and deterministic drivers of spatial and temporal turnover in breeding bird communities, Global Ecology and Biogeography 22:202-212. Because such an approach would be based on your empirical species and heterogeneity data, wouldn’t it serve as a more robust null model to your specific type of data? Notice, however, that I think that your approach is valid. I’m only worried that it mostly tests for a lack of sampling effects in richness – heterogeneity relationships in general (since you generated artificial environments and virtual species), and not in your specific dataset. Please address this issue in the discussion, and if you think there’s a problem in using the standard null-model approach for testing sampling effects, please discuss it as well.

Specific comments:

Line 82 – The first PCA axis explained only 32.5% of the variation in environmental conditions, but you used it as the sole source of heterogeneity. Please discuss how this might affect your results.

Line 92 – Any specific reason for using 75km?

Validity of the findings

Despite my minor reservations about the approach, I agree that the findings are valid. The conclusions are appropriate, and the authors do a good job of relating their findings to the broader context of biodiversity conservation.

Additional comments

Thank you for an interesting manuscript!

Reviewer 2 ·

Basic reporting

The manuscript seems to focus on testing something that is now considered common knowledge (i.e., habitat heterogeneity hypothesis) and it is not clear why the extra work is needed. It seems to me that this paper is stating something important to add to current knowledge (something along the variation with the magnitude of heterogeneity) but the scene is not well set and in the end the main result is the one we already knew from the first sentence of the Introduction. I believe this manuscript could be improved if the authors clarified the main aim and highlighted the differences in relation to previous knowledge.

Experimental design

The authors seem to have been careful with the methods design and assessment; however (and in relation to my previous point) it was not clear why this needed to be tested – as no new hypothesis seem to be set up. The “magnitude” related findings do not seem to be well presented (as methods to test specifically this are not clear).

Validity of the findings

Same as basic reporting.

Additional comments

Same as basic reporting and Experimental Design.

Also, title could be shortened to something like:
“Testing the influence of environmental heterogeneity on fish species richness” and perhaps focus on magnitude (?)
Starting sentences in abstract seem to merge two unrelated subjects: rapid loss of species with influence of habitat heterogeneity in richness patterns (?) Please clarify if inferring that these are directly related. Also, please clarify how and which avenues are open from this work.

---

## Round 0.2 · accepted · Accept

Tank you for your careful attention to the editors and referees commnets, and improvements to the paper.